# Chemerin as Potential Biomarker in Pediatric Diseases: A PRISMA-Compliant Study

**DOI:** 10.3390/biomedicines10030591

**Published:** 2022-03-03

**Authors:** Katarzyna Zdanowicz, Anna Bobrus-Chociej, Dariusz Marek Lebensztejn

**Affiliations:** Department of Pediatrics, Gastroenterology, Hepatology, Nutrition and Allergology, Medical University of Bialystok, 15-269 Bialystok, Poland; anna.chociej@op.pl (A.B.-C.); lebensztejn@hoga.pl (D.M.L.)

**Keywords:** chemerin, children, adolescents, pediatrics

## Abstract

Adipose tissue is the main source of adipokines and therefore serves not only as a storage organ, but also has an endocrine effect. Chemerin, produced mainly in adipocytes and liver, is a natural ligand for chemokine-like receptor 1 (CMKLR1), G-protein-coupled receptor 1 (GPR1) and C-C motif chemokine receptor-like 2 (CCRL2), which have been identified in many tissues and organs. The role of this protein is an active area of research, and recent analyses suggest that chemerin contributes to angiogenesis, adipogenesis, glucose homeostasis and energy metabolism. Many studies confirm that this molecule is associated with obesity in both children and adults. We conducted a systematic review of data from published studies evaluating chemerin in children with various disease entities. We searched PubMed to identify eligible studies published prior to February 2022. A total of 36 studies were selected for analysis after a detailed investigation, which was intended to leave only the research studies. Moreover, chemerin seems to play an important role in the development of cardiovascular and digestive diseases. The purpose of this review was to describe the latest advances in knowledge of the role of chemerin in the pathogenesis of various diseases from studies in pediatric patients. The mechanisms underlying the function of chemerin in various diseases in children are still being investigated, and growing evidence suggests that this adipokine may be a potential prognostic biomarker for a wide range of diseases.

## 1. Introduction

Currently, adipose tissue is considered not only a storage organ, but is also an endocrine organ due to the production of several cytokines. Adipokines are associated with many physiological and pathophysiological processes in the human body. One of them, chemerin, is expressed not only in adipose tissue, but also in the liver, female reproductive organs, adrenal glands, lungs, fibroblasts, chondrocytes, pancreas, kidney, epithelial cells and platelets [1]. The role of chemerin is an active area of investigation, and recent studies suggest the role of chemerin in angiogenesis, adipogenesis, glucose homeostasis and energy metabolism [2,3].

The proteolytic processing of chemerin and its receptors has been described in detail, and therefore, we only mentioned it in this review to understand the purpose of this molecule’s use in clinical studies in children and adolescents.

Chemerin was first described as tazarotene-induced gene 2 protein (TIG2) in patients with psoriasis; expression of this gene was intensified in patients treated with the antipsoriatic retinoic acid tazarotene [4]. Further studies revealed that the TIG2 gene product is a natural ligand for chemokine-like receptor 1 (CMKLR1, previously described as ChemR23) and, consequently, described by Wittamer et al. as a protein chemerin [5]. It is initially synthesized as an inactive 163-amino-acid proprecursor (preprochemerin) and, due to N-terminal truncation, forms a 143-amino-acid prochemerin. Prochemerin is activated through proteolytic processing to active chemerin due to C-terminal cleavage by a number of serine and cysteine proteases of the inflammatory, coagulation and fibrinolytic cascades [6]. As a result of proteolytic cleavage, various isoforms with an unclear role are formed. For example, plasmin-mediated chemerin cleavage forms inactive chemerin-K158 and then undergoes further cleavage by carboxypeptidases N to active chemerin-S157 [6,7]. Additionally, two different enzymes involved in coagulation cascades, factors Xa and VIIa, activate chemerin-S157. Chemerin-S157 may also be generated by elastase released from neutrophils. Mast cell chymase converts chemerin-S157 into inactive chemerine-F154 [8]. Prochemerin acts as a substrate to kallikrein 7, a skin-specific enzyme, and forms active chemerin-F165 [9]. In addition, prochemerin is cleaved by neutrophil-derived enzyme proteinase 3 into inactive chemerin-A155. This observation suggests a relationship between tissue damage, blood coagulation and the activation of chemerin [10]. Prochemerin circulates at a relatively higher concentration in blood plasma, and chemerin-S157 dominates in adipose tissue, where expression of elastase and tryptase is involved in the activation of prochemerin. Chemerin mRNA expression was observed in the liver, white adipose tissue and skin, mainly in the epidermis [11]. This protein was also detected in cartilage and synovial fluid, where it intensified the production of proinflammatory cytokines and participated in cartilage degeneration and ossification [12]. In animal models, chemerin mRNA was also expressed in the spleen, colon, testes and uterus [7].

Chemerin is a natural ligand for chemokine-like receptor 1 (CMKLR1), G-protein-coupled receptor 1 (GPR1) and C-C motif chemokine receptor-like 2 (CCRL2), which have been identified in multiple tissues and organs [13]. This molecule binds to CMKLR1 and GPR1 with the same affinity, but lower affinity to CCRL2 [14]. Expression of CMKLR1 has been reported in adipose tissue, cells of the immune system, lungs, placenta, heart and bone cells. CMKLR1, G-protein-coupled, signals via mitogen-activated protein kinase (MAPK), extracellular signal-regulated kinase (ERK) and phosphatidylinositol 3 kinase (PI3K)/AKT pathways to influence biological function. Chemerin/CMKLR1 binding promotes the chemotaxis of leukocyte populations to the tissues, where the inflammatory process takes place [14]. Through CMLKR1, chemerin can regulate the osteoblastogenic differentiation of mesenchymal stem cells [12]. Binding of chemerin to CMLKR1 resulted in the progression of proliferative diabetic retinopathy [15]. This molecule may also be involved in the regulation of reproduction. According to animal studies, the highest expression of CMKLR1 in the endometrium was observed during the period of placentation and implantation [16].

GPR1 is expressed in the central nervous system, skin and adipose tissue, as well as in a few cell types, including Leydig cells and granulosa cells. Similarly to CMKLR1, GPR1 activates ERK1/2-MAPK pathways and promotes RhoA/ROCK-dependent pathways. The role of this receptor is still unclear. In mice, GPR1 was involved in the reduction of insulin sensitivity and the development of glucose intolerance. However, in the same study, Rourke et al. observed no effect of GPR1 on body weight and adipose tissue mass [7,14,17]. According to the most recent data, GPR1 is a signaling receptor for arrestin-based signaling and a scavenger receptor with broader ligand specificity [18].

CCRL2 differs from the other two receptors and is most closely related to the atypical family of chemokine receptors. The main function of the membrane protein is to anchor chemerin in order to increase its local concentration. When CCRL2 binds to chemerin, signaling pathways, calcium mobilization or ligand internalization are not activated [19]. Moreover, chemerin receptors display selective signaling properties [14].

The wide distribution of both chemerin and its receptors may be responsible for its complex influence on the functioning of the human body. The mechanism of action of chemerin through various receptors is shown in Figure 1. The molecule is connected with the inflammatory process and whole-body energy homeostasis. Based on the literature, chemerin may be both a proinflammatory and anti-inflammatory molecule, depending on the enzymes involved in the chemerin cleavage, the target cell type and the receptors with which it binds [20]. In an animal model, chemerin promoted the adhesion of macrophages to extracellular matrix proteins and adhesion molecules, suggesting its involvement in enhancing the inflammatory response [21]. On the other hand, in the case of acute lung injury, this adipokine affects the reduction of the inflammatory process by reducing neutrophil infiltration and inflammatory cytokine release [22]. In the animal model, Jaworek et al. noticed that the infusion of chemerin, prior to induction of acute pancreatitis, reduced the histological symptoms of acute pancreatitis, decreased serum amylase activity and the concentration of tumor necrosis factor (TNF) α. This is probably due to a reduction in NF-κB signaling [23]. On the other hand, Szpakowicz et al. found a positive correlation between chemerin and inflammatory markers (white blood cell count, neutrophil to lymphocyte ratio, highly sensitive C-reactive protein (CRP)) in patients with chronic coronary syndrome [24].

Chemerin is also considered an important biomarker of benign and malignant tumors such as non-small-cell lung cancer [25]. Not only are the systemic effects are emphasized, but also the auto/paracrine role of the molecule produced by tumors or adipocytes in the vicinity of the tumor. Until now, the expression of chemerin has been assessed in many neoplasms, such as lung cancer, colorectal cancer, breast cancer, ovarian cancer, prostate cancer, melanoma or hepatocellular carcinoma [26].

The protein has multiple effects on the cardiovascular system, digestive, pulmonary, renal or reproductive systems [27]. In addition, various correlations between individual adipokines were noted. It has been shown, inter alia, that chemerin positively correlates with leptin and negatively with adiponectin [28]. In this review, we will briefly describe the role of chemerin in the pathogenesis of various diseases from studies conducted in pediatric patients (Figure 2).

## 2. Materials and Methods

A review of MEDLINE/PubMed data was carried out in February 2022, using the phrase ‘chemerin’ in combination with ’children’, ‘adolescents’ or ‘pediatrics’, returning 84, 66 and 65 records, respectively. A detailed analysis was performed, with particular emphasis on the study of the effect of chemerin on the pathogenesis of various diseases in the pediatric population. Screening of the titles and abstracts was independently made by two investigators. The case–control studies included in this analysis fulfilled inclusion criteria: (1) explored the effect of chemerin on development and diagnosis of pediatric disorders, (2) at least 10 patients included in the study group and (3) had available full texts. We excluded from the analysis duplicates and articles published over 8 years ago. After eliminating abstracts from conferences, letters to the editor, meta-analyses and reviews, 37 articles remained. Next, the selected papers were discussed with all authors. Following detailed research, which aimed to keep only the studies significant to the scope of this review, only 36 articles were selected (Figure 3). The PRISMA guidelines were followed [29]. This was a literature-based study; thus, no ethical approval was required.

## 3. Results

### 3.1. Chemerin and Obesity

Obesity is currently considered a global epidemic, and according to the latest data, the prevalence of obesity in the pediatric population is increasing [30]. Genetic and environmental factors are responsible for the development of obesity among children and adolescents. Monogenic obesity is a rare, severe and early-onset type of obesity associated with inappropriate eating behavior, in particular extreme hyperphagia and endocrine disorders [31]. However, in most cases, excess nutrients and a sedentary lifestyle are the most important factors in childhood obesity. Many other environmental factors such as traffic pollution, use of antibiotics or exposure to famine can also influence the development of obesity [30]. The chemerin released from white adipose tissue influences its homeostasis, inflammation and adipocyte metabolism. In obese patients, the main source of this protein is subcutaneous or visceral fat and possibly the liver. Increased expression of chemerin in adipocytes is stimulated by TNF and lipopolysaccharide, which may indicate an association of adipokine with the inflammatory process [1]. Chemerin is expressed especially in white adipose tissue and only a small amount in brown adipose tissue. Brown adipose tissue, producing heat instead of storing ATP, is a potential target in the treatment of obesity. An important factor in the thermogenesis of brown adipose tissue is retinoid acid associated with the expression of chemerin. This observation may indicate the role of chemerin in the activity of brown adipose tissue [27].

In adults, chemerin level is significantly higher in serum of the overweight/obese patients and correlates with body mass index (BMI) [27]. The predictor of adult obesity is childhood obesity. Obese adults who were obese in childhood have a worse prognosis for diabetes and cardiovascular disease, as well as psychological and social problems. Moreover, the low birth weight associated with rapid weight gain in infancy predisposes to metabolic syndrome in adulthood. Despite differences in obesity in pediatric and adult patients, similar trends in chemerin levels were observed in most studies [32]. Most studies conducted in children have found higher levels of the molecule in overweight/obese individuals than in lean children [33,34,35,36,37,38,39], and only one report found no differences in chemerin levels between children with overweight/obesity and normal BMI [40]. In patients with atrophy of adipose tissue in the course of anorexia nervosa, the concentration of chemerin was lower than in the group of obese and healthy peers [41]. Niklowitz et al. also found a significant association of chemerin with waist circumference, waist-to-height ratio and fat content based on bioimpedance analysis, but not with adipose tissue based on skinfold thickness [34]. This observation may suggest that subcutaneous fat is not involved in the production of chemerin.

The effect of this adipokine on lipid metabolism was also confirmed. Higher concentrations of the molecule were associated with increases in serum triglyceride (TG), total cholesterol and low-density lipoprotein cholesterol (LDL-C), as well as low levels of high-density lipoprotein cholesterol (HDL-C) [27,33,37,38,42]. Apart from lipid dysregulation, it plays a role in glucose metabolism and positively correlates with homeostatic model assessment for insulin resistance (HOMA-IR) [33,34,37,38,39]. There have been conflicting results regarding the link between chemerin and glucose. In one study, chemerin was negatively associated with fasting glucose [40], while in another, the correlation between two substances was assessed as positive. Chemerin may be useful to monitor adolescents with type 1 diabetes mellitus (T1DM). In a study group including 40 children with T1DM, a statistically significant increase in chemerin levels was observed compared to healthy counterparts. In the study group, chemerin positively correlated with urea, fasting blood glucose, glycosylated hemoglobin and albumin/creatinine ratio. However, the correlation of glomerular filtration rate (one of the parameters of kidney function) and chemerin was insignificant [43].

In obese children without metabolic syndrome components, chemerin levels also positively correlated with the inflammatory parameter, CRP. This observation may indicate that chemerin may be a useful marker of obesity-induced low-grade inflammation in prepubertal children. In this study, no differences were found in other acute-phase indicators (white blood cells, interleukin-6) between obese and normal groups [36]. A positive association between serum chemerin and a proinflammatory diet according to the Childhood Inflammation Index (C-DII ™) was also observed in school-aged children. C-DII ™ is a new tool for assessing inflammatory properties, and higher values are associated with cardiometabolic risk. In this study, C-DII ™ was not associated with central and total adipose tissue. However, a direct link between C-DII ™ and chemerin was observed, regardless of the child’s age, gender, race, per capita income, screen time and body fat [44].

In addition, obesity is a risk factor for lower 25 (OH) D values. However, intervention studies did not confirm a beneficial effect of vitamin D supplementation on weight reduction [45]. Reyman et al. in a study of obese children found a link between 25 (OH) D deficiency and increased levels of circulating chemerin. Hierarchical cluster analysis confirmed the overrepresentation of 25 (OH) D deficiency in obese pediatric patients expressing inflammatory mediator clusters with high levels of chemerin. The mechanism underlying the relationship between 25 (OH) D deficiency and high levels of chemerin is unclear and more research is needed [46].

A decrease in the level of the molecule was also observed after lifestyle changes (exercise, diet modification) in obese adolescents [34,36,47,48,49]. In a group of 88 obese patients during a year of training based on exercise, nutritional education and behavioral therapy, including individual psychological care for the child and their family, half of them lowered their BMI-standard deviation score (BMI-SDS). Changes in the level of chemerin were also observed in comparison with obese children with a stable or increased SDS. Weight loss was associated with a significant decrease in chemerin. Changes in adipokine values were significantly associated with changes in BMI, BMI-SDS, waist circumference, waist-to-height ratio and body fat based on bioimpedance analyses. It is worth noting that all of the children enrolled in the program had similar levels of chemerin at the beginning of the study [34]. In a study involving obese boys, the effects of six weeks of dark chocolate supplementation (30 g per day of dark chocolate containing 83% cocoa) combined with interval jump rope exercise on adipokines and body composition were assessed. According to the study protocol, participants were divided into four groups depending on the interventions undertaken (jump rope exercise + white chocolate supplementation; jump rope exercise + dark chocolate supplementation; dark chocolate supplementation and control). All three interventional studies significantly reduced body weight, waist-to-hip ratio and fat mass. However, only boys who exercised with a skipping rope and consumed dark chocolate had lower levels of chemerin after the intervention [47]. Liu et al. found that 4-week aerobic exercise and slight dieting reduced the level of serum chemerin in obese female adolescents. Following lifestyle interventions, decreased chemerin was found to be positively correlated with weight loss, total body fat, trunk fat, waist circumference, fasting plasma glucose, fasting insulin, HOMA-IR, TG and total cholesterol. However, in the dieting group with an inactive lifestyle, there were no changes in chemerin levels after the interventions [48]. In children and adolescents with abdominal obesity, after an interdisciplinary intervention involving the Mediterranean diet, physical activity and nutritional education, significant reductions in chemerin levels between baseline and measurement at Month 2 of maintenance were observed. Moreover, based on these studies obese adolescents should be recommended a properly balanced diet combined with physical activity.

Higher levels of chemerin and lipopolysaccharide-binding protein were associated with a greater number of metabolic syndrome components [49]. Additionally, in obese patients with suspected nonalcoholic fatty liver disease (NAFLD) after L-carnitine supplementation, a significant reduction in the level of chemerin was observed, as well as a decrease in BMI, waist circumference, hip circumference and waist/hip ratio, liver enzymes, fasting plasma glucose and insulin [33]. However, in another study, curcumin supplementation (500-mg curcumin per day along with a slight weight loss diet for 10 weeks) did not reduce the concentration of chemerin in overweight and obese girls [50].

### 3.2. Chemerin and Cardiovascular Diseases

The involvement of chemerin in cardiovascular disease is multifactorial. This adipokine induces matrix metalloproteinases (MMPs) and promotes vascular remodeling and angiogenesis [20]. In animal studies, chemerin induced pulmonary artery contraction [51]. Recent studies have demonstrated the key role of chemerin in the pathogenesis of adult cardiovascular disease, including coronary artery disease, cardiomyopathy, ischemic stroke and plaque instability, and its use as a prognostic marker for cardiovascular complications [20,52]. Chemerin may also protect against vascular calcification in adult patients with chronic kidney disease (CKD) [53].

Several studies on the role of chemerin in cardiovascular disease have been conducted in the pediatric population. In one of them, in a group of obese children and adolescents without complications of metabolic syndrome (hypertension, diabetes, impaired fasting glucose level, hypertriglyceridemia, low HDL), a significant correlation of circulating molecule levels with systolic blood pressure load in 24 h blood pressure monitoring (ABPM) was noted. Additionally, the authors detected higher systolic load in 24 h ABPM in obese children with elevated levels of chemerin. These results suggest a link between higher levels of chemerin and increased systolic blood pressure in obese children. However, in the same study, chemerin did not correlate with ultrasound markers of arterial stiffness such as common carotid arteries thickness and abdominal aorta intima-media thickness (AIMT) [54].

Girls with Turner syndrome often have an atherogenic profile of cardiovascular risk factors, including higher obesity rates, impaired glucose metabolism and an impaired lipid profile. In a study of girls with partial or complete absence of one X chromosome, higher levels of chemerin were observed in overweight/obese patients compared to patients with normal weight and healthy controls. Serum chemerin was significantly higher in girls with Turner syndrome without metabolic syndrome than in healthy controls, despite being matched for age, BMI Z-score and waist circumference. In addition, circulating protein positively correlated with epicardial and perihepatic adipose tissue thickness, which is a well-known predictor of cardiometabolic risk (as measured by cardiac magnetic resonance imaging) [55]. This observation may indicate that the increased risk of metabolic disorders in patients with Turner syndrome is independent of BMI.

Serum chemerin levels were also assessed in children with Kawasaki disease, where the main complications are related to the cardiovascular system (coronary artery abnormalities, myocarditis, pericarditis, pericardial effusion, valvular dysfunction, left ventricular dysfunction and arrhythmias). Recent studies have found significantly higher levels of chemerin in children diagnosed with Kawasaki disease than in patients with fever from other causes or in healthy children [56,57]. Zhang et al. found no correlation between serum chemerin and the presence of a lesion in the coronary artery in children with Kawasaki disease [56]. On the contrary, Xiang et al. noticed higher circulating chemerin in patients with coronary artery defects than in patients without coronary anomalies. Interestingly, in the group with increased levels of chemerin, a better response to intravenous immunoglobulin treatment was observed [57]. More research is needed to evaluate the usefulness of chemerin as a marker of Kawasaki disease severity.

### 3.3. Chemerin in Neonatal Studies

The assessment of the concentration of chemerin in the youngest age group concerned only small for gestational age (SGA) infants, which is a risk factor for metabolic alterations (type 2 diabetes, insulin resistance, arterial hypertension, obesity and dyslipidemia in adulthood). In the study of Léniz et al., the levels of chemerin did not differ significantly in children with SGA at 3 and 24 months of age. However, positive correlations were found between chemerin and TG, insulin, glucose and HOMA-IR. Interestingly, after adjusting for body weight, length and gestational age, only the chemerin–insulin relationship was maintained. Chemerin in the third month of life positively correlated with TG in the second year of life. This suggests the possibility of using this molecule as a prognostic factor for hypertriglyceridemia in later life [58].

In most infants with SGA, catch-up growth occurs in a spontaneous way from 6 months to 24 months after birth and normal body weight is revealed by the age of 24 months. However, normally developing children with SGA are at risk of occurrence metabolic syndrome in adulthood. Léniz et al. found higher levels of chemerin in SGA infants with slow catch-up than in the normal catch-up group. Interestingly, compared to other adipokines, chemerin levels remained stable throughout the first two years of life. There were also no gender differences in the concentrations of chemerin. These data may indicate the usefulness of using chemerin as a marker of SGA children’s development [59]. In another study conducted among adult pregnant women, no relationship was found between maternal chemerin concentration at various stages of pregnancy and neonatal anthropometry [60].

### 3.4. Chemerin and Hepatobiliary Diseases

There is evidence of the involvement of chemerin in the pathogenesis of hepatobiliary disease in children. The main area of research is NAFLD, which may be an indicator of metabolic syndrome [61]. According to the North American Society of Pediatric Gastroenterology, Hepatology and Nutrition (NASPGHAN), liver biopsy is a standard procedure in the diagnosis of NAFLD [62]. However, this procedure is invasive, it may be associated with the risk of complications (bleeding, pain, accidental damage to a nearby organ), and in children, it is also associated with general anesthesia. Less-invasive biomarkers are needed to predict the presence and severity of NAFLD and to eliminate alternative diagnoses.

Chemerin levels are significantly elevated in overweight and obese children with NAFLD [33,63,64]. Hamza et al. found higher levels of chemerin in patients with elevated levels of liver enzymes [33]. Kłusek-Oksiuta et al. revealed the link between the concentration of adipokine, gamma glutamyltransferase and severity of liver steatosis in imaging studies (intensity of the hepatic steatosis in ultrasonography and intrahepatic lipid content in spectroscopy). According to this study, chemerin can be used as a predictor of NAFLD, and it is advanced in obese children [63]. On the other hand, no statistically significant difference in chemerin levels was observed between pediatric patients depending on specific histological changes such as lobular inflammation, ballooning or fibrosis grades. However, in the same study, the 186.7 ng/mL cut-off value for chemerin with almost 90% specificity can differentiate patients with hepatic steatosis from healthy children [59]. Positive correlations of systemic chemerin with ALT activity have also been described in obese children without chronic liver disease [39].

Increased levels of chemerin have been observed in children and adolescents with cholelithiasis. Although obesity is a well-known trigger of gallstones, higher levels of chemerin have been observed not only in obese, but also in lean children with gallstones compared to healthy children. In this study, a positive correlation between chemerin and TG was also noted [65]. These data suggest a potential role for this molecule in the development of gallstone disease in the pediatric population.

### 3.5. Chemerin in Other Diseases in Children

The role of chemerin in the pathophysiology of allergy [66], epilepsy [67], lung [68] or kidney diseases [69] remains largely unexplored and to the best of our knowledge, only single studies are available. In children with cow’s milk allergy, a significantly higher serum concentration of this molecule was demonstrated than in healthy subjects with a positive correlation between chemerin and CRP levels. In addition, there were no differences in the value of chemerin in various allergy mechanisms (IgE-mediated and non-IgE-mediated), clinical symptoms or the duration of the diet. Comparable levels of this molecule have been observed in subgroups of allergic patients with different clinical symptoms related to skin lesions, respiratory and gastrointestinal symptoms. There was no significant difference in this biomarker between subgroups with different diet duration (6–12 months versus 13–24 months versus >24 months) [66].

Elevated chemerin values may serve as a biomarker of hypoxia. Hypoxia activates the cascade of proinflammatory cytokines. Simultaneous changes in the concentration of chemerin were also observed in children with idiopathic epilepsy. Higher levels of chemerin have been reported in patients with more severe epilepsy, especially with poor seizure control. This study showed a positive correlation between serum protein levels and the severity of seizures and the duration of epilepsy. According to Elhady et al., it may be a predictor of drug response in idiopathic childhood epilepsy [67].

Cystic fibrosis is a chronic disease with persistent, high-intensity inflammation. Malnutrition in patients with cystic fibrosis is often a problem associated with lung deterioration, reduced exercise tolerance, growth retardation and reduced quality of life. Dietary deficiencies in cystic fibrosis are associated with gastrointestinal complications such as nutritional deficiencies, including fat and fat-soluble vitamins, diabetes, liver dysfunction and cholelithiasis [70]. In pediatric cystic fibrosis patients, chemerin levels were comparable to healthy controls and did not correlate with nutritional status and levels of interleukin-1 beta (IL-1b), interleukin-6 (IL-6) and TNF α [68].

CKD is a risk factor for cardiovascular disorders in both children and adults. In a study of children with CKD, Szczepańska et al. showed a lower level of chemerin compared to the healthy anthropometrically matched control group. The concentration of chemerin was negatively correlated with body weight. This relationship may result from nutritional limitations or poor appetite in children with CKD. The values of this adipokine were comparable in girls and boys. Additionally, the type of renal replacement therapy (hemodialysis vs. peritoneal dialysis) had no effect on the concentration of chemerin [69].

Despite the lack of published studies on chemerin in sepsis in children, interesting data come from studies conducted among adult patients. Circulating chemerin is increased in the early stages of sepsis, especially in patients with septic shock, according to recently published data. Moreover, correlations between chemerin and white blood cells, lactates, CRP and procalcitonin as well as biomarkers of glucose homeostasis were observed [21].

The concentrations of chemerin reported in children based on selected studies are shown in Table 1.

## 4. Conclusions

Accumulating evidence points to chemerin as a potential prognostic biomarker for a wide range of diseases. However, the mechanism that regulates the concentration of chemerin is still not clearly described. In addition, the development of therapeutic options that will act on chemerin and its receptors appears to be beneficial in the treatment of obesity and metabolic syndrome. More research is needed to elucidate the use of chemerin in daily clinical practice. Based on an increasing amount of research, the effect on chemerin or its receptors may serve as a new therapeutic approach to the treatment of not only obesity-related diseases.

## Figures and Tables

**Figure 1 biomedicines-10-00591-f001:**
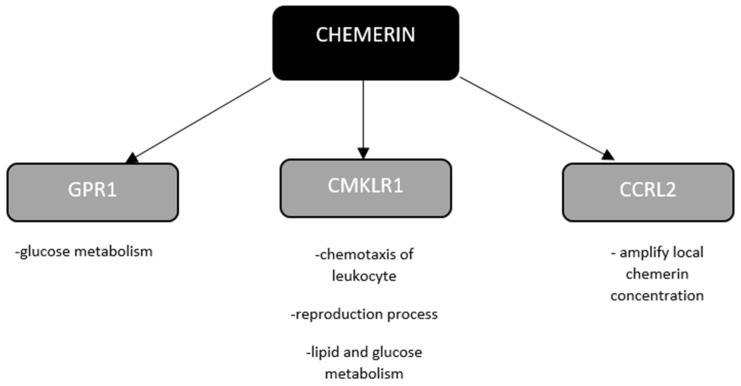
Mechanism of chemerin. Chemokine-like receptor 1 (CMKLR1), G-protein-coupled receptor 1 (gPR1) and C-C motif chemokine receptor-like 2 (CCRL2).

**Figure 2 biomedicines-10-00591-f002:**
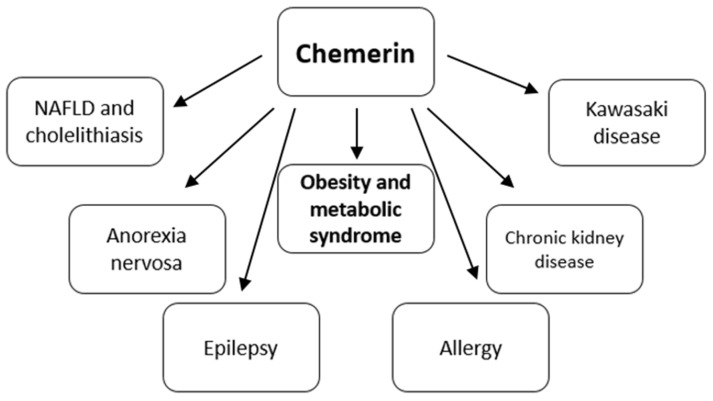
Involvement of chemerin in various diseases in children and adolescents.

**Figure 3 biomedicines-10-00591-f003:**
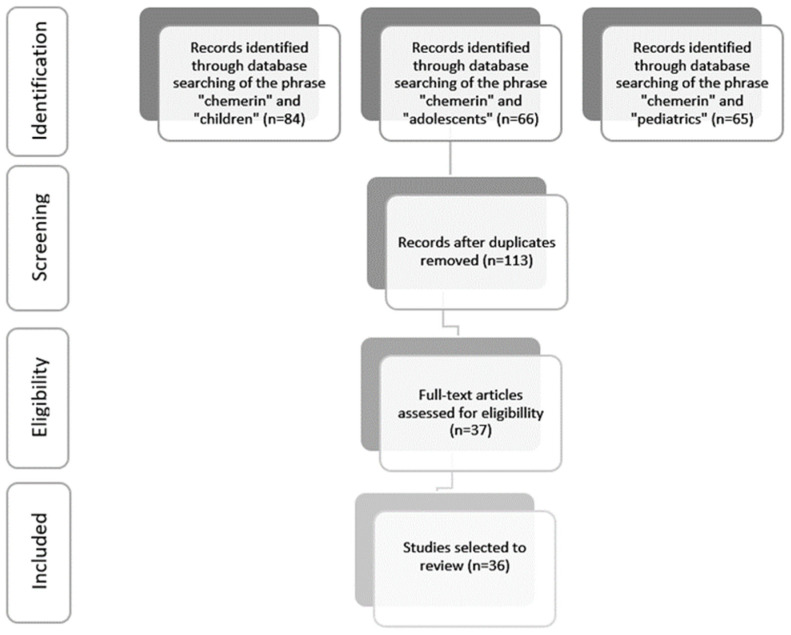
Flow of information through the different phases of the systematic review.

**Table 1 biomedicines-10-00591-t001:** Chemerin concentrations reported in children based on selected research studies.

Authors	Analyzed Population	Number of Included Patients (*n*)	Chemerin Concentration
Hamza RT et al. [33]	Obesity	50	Increased
Niklowitz P et al. [34]	Obesity	88	Increased
Salem DA et al. [38]	Toxoplasma gondiiseropositive-obese group	28	Increased
Oświecimska et al. [41]	Anorexia nervosa	65	Decreased
Elsehmawy AAEW et al. [43]	T1DM	40	Increased
Reyman M et al. [46]	25(OH)D deficient, obesity	36	Increased
Salem NA et al. [55]	Turner syndrome	46	Increased
Zhang XY et al. [56]	Kawasaki diasease	80	Increased
Léniz A et al. [59]	SGA slow vs. normal catch-up	27	Increased in group with slow catch-up
Kłusek-Oksiuta M et al. [63]	NAFLD	45	Increased
Zdanowicz K et al. [65]	Cholelithiasis	54	Increased
Ambroszkiewicz J et al. [66]	Allergy	64	Increased
Elhady M et al. [67]	Epilepsy	50	Increased
Sznurkowska K et al. [68]	Cystic fibrosis	72	Unchanged
Szczepańska M et al. [69]	CKD	28	Decreased

T1DM—type 1 diabetes mellitus, SGA—small for gestational age, NAFLD—nonalcoholic fatty liver disease CKD—chronic kidney disease.

## Data Availability

Not applicable.

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
