# Peer review of "Chemerin as Potential Biomarker in Pediatric Diseases: A PRISMA-Compliant Study"

_biomedicines, 2022, doi:10.3390/biomedicines10030591_

Round 1
Reviewer 1 Report
Dear authors,
This is a systematic review of clinical studies regarding the role of chemerin in the pathogenesis of various diseases in the pediatric population. The topic is very interesting since chemerin is a novel adipokine that has not been extensively studied. You have summarized nicely the most important findings from relative studies. Your review is comprehensive and informative to the reader. However, there are some points that you would like to consider in order to improve your manuscript:
- The aim of the study, as well as the design (systematic review) and the criteria used to select the reported studies should be reported in the abstract. You may also report how many studies you included and what kind of studies supports your findings.
- Line 27: Many other tissues and cells express chemerin: the skin, the pancreas, the kidney, fibroblasts, and epithelial cells.
- Lines 88-93: Please explain in more detail the multifaceted role of chemerin in immune regulation (pro- and anti-inflammatory roles).
- Lines 101-106: Please add some more text on the role of chemerin in lung cancer. Suggested reference: doi: 10.1016/j.lungcan.2018.10.010,
- Line 185: Please add a reference supporting low 25(OH) levels in obesity, for example doi: 10.1007/s13679-021-00433-1
- Line 324: 3.5. Chemerin in other diseases in children : It would be useful to add a short paragraph regarding the role of chemerin in sepsis. There is a recently published study on serum chemerin in adult patients with sepsis, supporting a role of chemerin as an emerging diagnostic and prognostic biomarker in sepsis (doi: 10.3390/biom12020301). Are there any studies on chemerin in neonatal sepsis, or in children?
Author Response
Thank you for your valuable comments. Changes in the manuscript are marked in yellow.
In response to a message from the first Reviewer:
- The abstract has been supplemented according to the Reviewer's recommendations.
- The information has been completed.
- The information has been completed.
- Information has been supplemented and a suggested publication has been quoted.
- Information has been supplemented and a suggested publication has been quoted.
- Information has been supplemented and a suggested publication has been quoted.
Reviewer 2 Report
It is necessary to include references that support the role of chemerin in pediatric diseases. Some concerns should be addressed before publication.
- Title: Be sure the title includes any specific terms as directed in the reporting guidelines for your type of article (for example, "case report" should be in the title of a CARE-compliant article). The following guidelines specify terms that should be in the title: CARE, CHEERS, CONSORT, PRISMA. Please modify the title according to the figure2 (with PRISMA). For example, PMID: 35146807.
- Why the authors didn’t conduct a meta-analysis for effect of chemerin in pediatric diseases after all reviewing relevant literatures.
- Search strategy didn’t be stated in text. The authors should address this concern.
- Could the authors add some mechanism plot of chemerin in pediatric diseases to meet the merit of Biomedicnes.
- Some references should be updated.
Author Response
Thank you for your valuable comments. Changes in the manuscript are marked in yellow.
In response to a message from the second Reviewer:
- The title has been changed as recommended.
- The meta-analysis was not carried out due to single studies cited in most diseases.
- The search strategy was described.
- Figure 1 has been added to the manuscript, which gives an overview of the mechanism of action of chemerin through its receptors.
- New citations have been added such as: 21, 25, 45.
Reviewer 3 Report
The authors reviewed the latest advances in knowledge of the role of chemerin in the pathogenesis of various diseases from studies in pediatric patients. A review of MEDLINE/PubMed data were carried out. A detailed analysis was performed, with particular emphasis on the study of the effect of chemerin on the pathogenesis of various diseases in the pediatric population. 36 articles were selected and evaluated. Based on their results, accumulating evidence points to chemerin as a potential prognostic biomarker for a wide range of diseases. However, the mechanism that regulates the concentration of chemerin is still not clearly described. In addition, the development of therapeutic options that will act on chemerin and its receptors appears to be beneficial in the treatment of obesity and metabolic syndrome. More research is needed to elucidate the use of chemerin in daily clinical practice.
Comments:
The review is informative, well written, and very interesting, particularly to clinicians.
- The abstract could be more informative. A brief conclusion of the review could be added.
- A figure demonstrating the chemerin’s mechanism of action could be improve the quality of the review.
- In Table 1., “Not significant” could be changed to “Unchanged”.
- There are many excellent papers focusing on the regulatory role of chemerin in adults. The authors should highlight the differences between an adult and pediatric populations.
- There are some typos throughout the manuscript.
- English needs some editing.
Author Response
Thank you for your valuable comments. Changes in the manuscript are marked in yellow.
In response to a message from the third Reviewer:
- The abstract has been supplemented according to the Reviewer's recommendations.
- Figure 1 has been added to the manuscript, which gives an overview of the mechanism of action of chemerin through its receptors.
- “Not significant” has been replaced with "Unchanged".
- The section on obesity highlights the differences between the child and adult populations, and highlights the complexity of childhood obesity and its impact on later life.
- English and typos were corrected by a native speaker.
Round 2
Reviewer 2 Report
No further comments. Thanks for your efforts on revision.